# Genome-Scale Metabolic Modeling with Protein Expressions of Normal and Cancerous Colorectal Tissues for Oncogene Inference

**DOI:** 10.3390/metabo10010016

**Published:** 2019-12-25

**Authors:** Feng-Sheng Wang, Wu-Hsiung Wu, Wei-Shiang Hsiu, Yan-Jun Liu, Kuan-Wei Chuang

**Affiliations:** Department of Chemical Engineering, National Chung Cheng University, Chiayi 62102, Taiwan; wwh@cs.ccu.edu.tw (W.-H.W.); math30512@gmail.com (W.-S.H.); kid5911027@gmail.com (Y.-J.L.); tomjuicf@gmail.com (K.-W.C.)

**Keywords:** oncogene, flux balance analysis, cancer cell metabolism, tissue-specific metabolic models, constraint-based modeling, multi-level optimization

## Abstract

Although cancer has historically been regarded as a cell proliferation disorder, it has recently been considered a metabolic disease. The first discovery of metabolic alterations in cancer cells refers to Otto Warburg’s observations. Cancer metabolism results in alterations in metabolic fluxes that are evident in cancer cells compared with most normal tissue cells. This study applied protein expressions of normal and cancer cells to reconstruct two tissue-specific genome-scale metabolic models. Both models were employed in a tri-level optimization framework to infer oncogenes. Moreover, this study also introduced enzyme pseudo-coding numbers in the gene association expression to avoid performing posterior decision-making that is necessary for the reaction-based method. Colorectal cancer (CRC) was the topic of this case study, and 20 top-ranked oncogenes were determined. Notably, these dysregulated genes were involved in various metabolic subsystems and compartments. We found that the average similarity ratio for each dysregulation is higher than 98%, and the extent of similarity for flux changes is higher than 93%. On the basis of surveys of PubMed and GeneCards, these oncogenes were also investigated in various carcinomas and diseases. Most dysregulated genes connect to *catalase* that acts as a hub and connects protein signaling pathways, such as those involving *TP53*, *mTOR*, *AKT1*, *MAPK1*, *EGFR*, *MYC*, *CDK8*, and *RAS* family.

## 1. Introduction

Recent developments in omics data, such as genomics [1], transcriptomics [2], proteomics [3], metabolomics [4], and fluxomics [5], can be observed by consulting numerous biological databases. Such advancement in high-throughput data acquisition has shifted the focus from data generation to processing and understanding how to integrate the collected information. Because the amount of available omics data is increasing rapidly, advanced computational methods are required to mine the relevant knowledge from these data sources to boost the development of genome-scale metabolic models (GSMMs) of several organisms, including humans. Metabolism is the primary biological mechanism that is linked from genotype to phenotype; it can help us understand cell physiology and certain disease phenotypes caused by metabolic dysregulation [6,7]. GSMMs composed of metabolites and reactions allow the representation of the full set of metabolic processes within a cell to be curated using knowledge of cellular functions from the literature. GSMMs combined with constraint-based approaches lead to the development of methods to simulate cell behavior, such as flux balance analysis (FBA) [8].

A better understanding of the genome-scale human metabolic network may enable the identification of disease genes and related pathways, thereby offering better targets for drug development. The release of genome-scale human metabolic networks, such as Recon 1 and 2 [9,10], Edinburgh human metabolic network (EHMN) [11], and human metabolic reactions [12], has led to the emergence of network medicine. Recon 2.2 and Recon 3D are the most comprehensive human genome-scale network reconstructions [13,14]. Recon 3D includes three-dimensional metabolite and protein structure data and enables an integrated analysis of metabolic functions in humans [13]. Recon 2.2 ensures the charge balance in all reactions and improves the simulation of energy metabolism [14].

Human tissues have diverse metabolic functions, which are complex and specialized in different tissues and cell types. Mapping out these tissue-specific metabolisms in GSMMs will advance our understanding of the metabolic basis of various physiological and pathological processes. Nevertheless, for the in-depth study of specific cell phenotypes, the development of tissue- or cell-type-specific metabolic models is necessary. Automated reconstruction algorithms developed to date can be broadly categorized into flux-dependent and pruning methods [15,16,17,18,19,20,21,22]. Flux-dependent methods identify an optimal genome-scale metabolic network (GSMN) and include the maximum number of high confidence reactions supported by substantial experimental data. In contrast, pruning methods start with a core set of reactions obtained through literature reviews or experimental data and proceed by removing the remaining reactions in the general reconstruction while maintaining functionality in the core set. Nevertheless, both algorithms aim to keep the final tissue-specific reconstruction as concise as possible. Schultz and Qutub [23] recently introduced a cost optimization reaction dependency assessment (CORDA) algorithm to build concise, but not minimalistic, tissue-specific metabolic models based on omics data and a general human metabolic reconstruction. Many biological databases, such as HPA [3], HMDB [4], and TCGA [24], are available and enable the reconstruction of tissue-specific metabolic models. Yizhak et al. have reviewed the challenges regarding to genome-scale modeling of cancer metabolism, and provided valuable insights for clinically relevant applications [25]. In this study, we first established an algorithm by integrating the CORDA method [23] with the HPA database [3] and the human metabolic network Recon 2.2 [14] to reconstruct GSMMs both healthy and cancerous colorectal tissues. Colorectal cancer (CRC), a malignant tumor of the large intestine with an incidence of approximately 10%, is the second and third most commonly occurring cancer in females and males, respectively. An estimated 50,630 deaths from CRC were reported in 2018 in the United States [26]. The Taiwan Cancer Registry report revealed that CRC was the most frequently diagnosed cancer in men and the second most frequently diagnosed cancer in women in 2017 [27]. Molecular biologists have discovered several genes that contribute to the susceptibility of the two types of colon cancer, namely familial adenomatous polyposis (FAP) and hereditary nonpolyposis colon cancer (HNPCC). Notably, genes causing HNPCC and FAP are relatively easy to discover, unlike those that cause susceptibility to CRC. In this study, we introduced a tri-level optimization framework to infer details on dysregulated genes that induce CRC through metabolic reprogramming in cancer cells. The GSMMs for the cancerous and healthy cells of the colorectal tissue were used to build templates of flux alterations between cells. The tri-level optimization problem is difficult to solve; therefore, the NHDE algorithm was applied to solve this problem.

## 2. Materials and Methods

### 2.1. Reconstruction of GSMNs

The metabolic network reconstruction processes are described in Figure 1. The process has seven steps and involves the integration of four databases to establish constraint-based models of cancerous and healthy tissue cells. Protein expression data for all genes that appeared in the gene association of Recon 2.2 were retrieved from the HPA database (Figure 1A). All reactions were classified into high, medium, and negative confidence groups based on the dependency assessment (Figure 1B). Exchange reactions for required nutrition uptake and reactions of well-known cancer pathways were involved in the high confidence group (Figure 1B). Recon 2.2, a human general metabolic network, and data on the dependence of tissue-specific reactions were provided as the input information for the CORDA algorithm to construct a tissue-specific metabolic network (Figure 1C), which was saved in Systems Biology Makeup Language (SBML) format (Figure 1D) for model exchange. Both cancer (CA) and healthy (HT) models were built by using the CORDA algorithm and merged into a basal (BL) model. Each tissue-specific metabolic network comprised thousands reactions and species, rendering it difficult to modify and analyze manually. We developed a systems biology program (SBP) platform that supported the SBML file to perform the simulation and analysis of metabolic networks in the general algebraic modeling system (GAMS) environment. The SBP platform can transfer a metabolic network in SBML format to its GAMS model automatically (Figure 1E). The BL model in Figure 1F was used to investigate how healthy cells undergo metabolic reprogramming to become cancer cells. Finally, we can employ the BL model to compute network topology and conduct a flux variability analysis (Figure 1G).

### 2.2. Flux Balance Analysis

FBA is a constraint-based modeling approach where the stoichiometry of the underlying biochemical network constrains the solution. The stoichiometric matrix of a typical metabolic system is underdetermined, and infinite solutions are possible. The FBA formulates a metabolic network as a linear programming problem, as shown in Equation (Equation 1), wherein the solution of the underdetermined system is a member of the solution space and optimizes an objective function of choice, such as maximal growth and maximal ATP synthesis rate.
(1)$$\left\{\begin{array}{c} \max_{v_{f/b}}\left(w_{ATP}v_{ATP}+w_{biomass}v_{biomass}\right)\\ \mathrm{subject}\mathrm{to}\\ \;N\left(v_{f}-v_{b}\right)=0\\ \;v_{f/b,i}^{LB}\leq v_{i}\leq v_{f/b,i}^{UB},i\in \Omega \end{array}\right.$$
where $v_{f/b}$ is the forward/backward flux vector of reversible reactions; N is an m×n stoichiometric matrix where *m* is the number of metabolites and *n* is the number of reactions; $v_{f/b,i}^{LB}$ and $v_{f/b,i}^{UB}$ are the positive lower and upper bounds of the $i^{th}$ backward/forward flux, respectively; Ω is the set of forward and backward fluxes. The coefficients ($w_{ATP}$ and $w_{biomass}$) are the weighting factors for computing the linear objective function. The objective of FBA is to maximize the biomass formation rate ($v_{biomass}$) for the cancer model. However, normal cells may have different objectives depending on growth signals [28]; thus, the maximum ATP synthesis rate ($v_{ATP}$) is applied in this situation.

FBA assumes that metabolic networks will reach a steady state constrained by the stoichiometry. Even though the stoichiometric constraints lead to an underdetermined system, the bounded solution space of all feasible fluxes can be identified, that is, a large set of alternative flux distributions with an identical objective value. We minimized the squared sum of all internal fluxes for FBA to ensure efficient channeling of all fluxes through all pathways to eliminate numerous flux distributions in Equation (Equation 1). The Euclidean norm problem for minimization is expressed as follows:(2)$$\left\{\begin{array}{c} \min_{v_{f/b}}\sum_{i\in \Omega^{Int}}{\left(v_{f,i}\right)}^{2}+{\left(v_{b,i}\right)}^{2}\\ \mathrm{subject}\mathrm{to}\\ \;N\left(v_{f}-v_{b}\right)=0\\ \;v_{f/b,j}^{LB}\leq v_{f/b,j}\leq v_{f/b,j}^{UB},j\in \Omega \\ \;w_{ATP}v_{ATP}+w_{biomass}v_{biomass}\geq obj^{*} \end{array}\right.$$
where $obj^{*}$ is the optimal objective value obtained from Equation (Equation 1), and $\Omega^{Int}$ is the set of intracellular reactions. The problem expressed in Equation (Equation 2) is a quadratic programming problem that has a unique solution.

### 2.3. Oncogene Inference Problem

We introduced a tri-level optimization problem (TLOP) to simulate gene screening procedures in a wet lab to infer oncogenes. The flowchart of the in silico experiment is presented in Figure 2. GSMMs of cancerous and normal cells were reconstructed, as shown in Figure 2A, the procedures of which are discussed in Figure 1. Both models were then applied to compute the flux distribution patterns at the cancer and normal situations (Figure 2B). The flux template, which acted as a control, was built according to the flux distributions of cancer and normal models (Figure 2C). Flux alterations were computed from each mutant (Figure 2D) and then compared with the control (Figure 2E). The template was incorporated with the TLOP to infer oncogenes, as shown in Figure 2E–I. The outer optimization problem was employed to decide which genes were modulated (Figure 2F). The mutated genes were provided for the inner optimization problem to compute the flux distribution pattern to evaluate the flux alteration (Figure 2D–G). The TLOP framework can be formulated by using the procedures from Figure 2C–I, and is expressed as follows:
(3)Outeroptimizationproblem:Fuzzyequaloflogarithmicfoldchangetothetemplate:equalmaxδ,z˜LFCmMUBL≈LFCmCABLequalmaxδ,z˜LFCmMUHT≈LFCmCAHTSimilarityratioofflux-sumsynthesisformetabolites:maxδ,zSRT,HT,maxδ,zSRT,BLSimilarityratiooffluxforreactions:maxδ,zSRTrxn,HT,maxδ,zSRTrxn,BLsubjecttotheinneroptimizationproblems:FBAproblemmaxvf/bwATPvATP+wbiomassvbiomasssubjecttoNBLvf−vb=0vf/b,iLB,MU≤vf/b,i≤vf/b,iUB,MU,zi∈ΩMUvf/b,jLB≤vf/b,j≤vf/b,jUB,zj∉ΩMUUFDproblemminvf/b∑i∈ΩIntvf,i2+vb,i2subjecttoNBLvf−vb=0vf/b,iLB,MU≤vf/b,i≤vf/b,iUB,MU,zi∈ΩMUvf/b,jLB≤vf/b,j≤vf/b,jUB,zj∉ΩMUwATPvATP+wbiomassvbiomass≥wATPvATP*+wbiomassvbiomass*
where $\tilde{\mathrm{equal}}$ is the fuzzy equal objective function that represent the fuzzy goals. For example, the $LFC_{m}^{MUBL}$ and $LFC_{m}^{CABL}$ should be restored to a state that is as close as possible; $N^{BL}$ is the stoichiometric matrix of basal models; the integer vector z is used to determine mutated enzymes; δ is the regulated strength parameter for the mutants; and $\Omega^{MU}$ is the set of mutated reactions.

The hierarchical multiple objectives are considered in the outer optimization problem in Equation (Equation 3). The first priority is to employ the fuzzy equal measure to determine that the mutant logarithmic flux changes $(LFC_{m}^{MUBL}$ and $LFC_{m}^{MUHT})$ are as close to the templates $(LFC_{m}^{CABL}$ and $LFC_{m}^{CAHT})$ as possible. The logarithmic fold change $\left(LFC_{m}\right)$ between the synthesis rates or fluxes of the $m^{th}$ metabolite in cancerous or dysregulated (denoted as deficient) and basal or healthy (denoted as normal) states is computed as follows:(4)$$LFC_{m}=log_{2}\left(\frac{r_{m,deficient}}{r_{m,normal}}\right)$$
where the overall synthesis rate ($r_{m}$) of the $m^{th}$ intracellular compound in deficient and normal states is evaluated as follows:(5)$$r_{m}=\sum_{i\in \Omega^{c}}\left(\sum_{N_{ij}>0,j}N_{ij}v_{f,j}-\sum_{N_{ij}<0,j}N_{ij}v_{b,j}\right),m\in \Omega^{m}$$

Here $\Omega^{c}$ is the set of metabolites located in different compartments, and $\Omega^{m}$ is the set of metabolites. The expression enclosed by brackets in Equation (Equation 5) indicates the synthesis rate of the $i^{th}$ metabolite that summed the influxes of the forward reactions and backward reactions. Each intracellular metabolite may exist in different compartments of the metabolic network, e.g., nine compartments in Recon 2.2, so that the overall synthesis rate of the $m^{th}$ compound is computed by Equation (Equation 5).

The second and third objectives are determining that the similarity ratios of the flux reprogramming are maximized. The similarity ratios $(SR^{T,BL/HT}$ and $SR^{T_{rxn},BL/HT})$ of the flux-sum synthesis for metabolites and ratios of fluxes for all reactions are evaluated as follows:(6)$$SR^{T/T_{rxn}}=\frac{\sum_{m=1}\left|\mu_{m}^{T/T_{rxn}}\right|}{N_{T/T_{rxn}}}$$
where the similarity indicator $\left(\mu_{m}^{T/T_{rxn}}\right)$ for the $m^{th}$ metabolite is defined as follows:(7)$$\mu_{m}^{T/T_{rxn}}=\left\{\begin{array}{cc} 1, & \mathrm{if}\;LFC_{m}>tol_{+}\mathrm{and}LFC_{m}^{T/T_{rxn}}>tol_{+}\\ -1, & \mathrm{if}\;LFC_{m}<tol_{-}\mathrm{and}LFC_{m}^{T/T_{rxn}}<tol_{-}\\ 0, & \mathrm{otherwise}. \end{array}\right.$$
where the logarithmic fold change $\left(LFC_{m}^{T/T_{rxn}}\right)$ of the $m^{th}$ metabolite for the templates are provided in advance. The tolerances for increase or decrease are defined as $tol_{+}=log_{2}\left(1+\varepsilon \right)$ and $tol_{-}=log_{2}\left(1-\varepsilon \right)$, respectively, and ε is the percentage of flux alteration. A numerical example is provided (Appendix A) to illustrate the computation of flux template, similarity ratio, and logarithmic fold change.

### 2.4. Association of Gene-Protein-Reaction

The TLOP framework must decide which genes are selected for mutation. Most optimal strain design methods [29,30,31,32,33] involve directly applying the reactions of the stoichiometric matrix to determine the optimal modulations. Posterior decision -making must be conducted to determine the corresponding gene that each optimal reaction integrates with the gene association. Such decision-making is tedious and incapable of identifying the reactions that are regulated by isozymes or redundant genes. A regulatory FBA framework [34] has been used to apply gene association incorporated with the stoichiometric matrix to formulate a connective matrix, wherein logical operations “AND” and “OR” were applied to build a gene-protein-reaction (GPR) expression, as shown in Figure 3A, for conceptual description. Nonetheless, such a GPR model is still unable to discriminate redundant genes, and the regulatory FBA becomes a discrete nonlinear optimization problem. This study introduced a strategy for enzyme pseudo-coding numbers to separate the original gene association into two parts, as shown in Figure 3B. The first part can identify the reductant pseudo-enzymes and isozymes, such that the reduced association catalyzes the reactions. The second part represents single gene or complex gene corresponding to an enzyme pseudo-coding number.

The pseudo-enzymes are then used as upper-level variables in Equation (Equation 3) to select modulated genes and the value for the regulated bounds is computed using the following equations:(8)$$\begin{array}{c} \mathrm{Upregulayion}:\\ \left\{\begin{array}{c} \left(1-\delta \right)v_{f,i}^{basal}+\delta v_{f,i}^{UB}\leq v_{f,i}\leq v_{f,i}^{UB}\\ v_{b,i}^{LB}\leq v_{b,i}\leq \left(1-\delta \right)v_{b,i}^{basal}+\delta v_{b,i}^{LB},i\in \Omega^{MU} \end{array}\right.\\ \mathrm{Downregulation}:\\ \left\{\begin{array}{c} v_{f,i}^{LB}\leq v_{f,i}\leq \left(1-\delta \right)v_{f,i}^{basal}+\delta v_{f,i}^{LB}\\ \left(1-\delta \right)v_{b,i}^{basal}+\delta v_{b,i}^{UB}\leq v_{b,i}\leq v_{b,i}^{UB},i\in \Omega^{MU}\setminus \Omega^{IZ} \end{array}\right.\\ \left\{\begin{array}{c} v_{f,i}^{LB}\leq v_{f,i}\leq v_{f,i}^{UB}\\ v_{b,i}^{LB}\leq v_{b,i}\leq v_{b,i}^{UB},i\in \Omega^{MU}\cap \Omega^{IZ} \end{array}\right.\\ \mathrm{Knockout}:\\ \left\{\begin{array}{c} v_{f,i}=0\\ v_{b,i}=0,i\in \Omega^{MU}\setminus \Omega^{IZ} \end{array}\right.\\ \left\{\begin{array}{c} v_{f,i}^{LB}\leq v_{f,i}\leq v_{f,i}^{UB}\\ v_{b,i}^{LB}\leq v_{b,i}\leq v_{b,i}^{UB},i\in \Omega^{MU}\cap \Omega^{IZ} \end{array}\right. \end{array}$$
where $\Omega^{IZ}$ is the set of reactions regulated by isozymes. A bound for down-regulation or knockout is assigned to its original region if the reaction is catalyzed by isozymes.

### 2.5. Nested Hybrid Differential Evolution Algorithm

The candidate genes in Equation (Equation 3) are represented by integer variables such that it is a mixed-integer optimization problem that is NP-hard [35]. Classical algorithms for solving bi-level optimization problems have applied duality theory to convert the inner level optimization problem into constraints in the outer-level problem. However, the duality transformation is difficult for multilevel optimization problems, such as TLOP in this study. In this study, we extended the NHDE algorithm to solve the TLOP [36]. The computational concept of NHDE (Appendix A is based on a hybrid differential evolution (HDE), which was extended from the original differential evolution (DE) algorithm. The basic operations of the NHDE algorithm are similar to those of the DE and HDE algorithms, except for coding representation, selection, and evaluation operations. The NHDE algorithm has been applied to solve metabolic engineering [33] and biomedical problems [36,37]. In the outer optimization problem, the NHDE algorithm is used to determine which pseudo-enzymes are selected to be modulated, and the inner optimization problems (FBA and UFD) are then solved using a linear and quadratic optimization solver. An optimal solution for each candidate individual is achieved when the UFD problem is convergent, and the set of these individual solutions comprises a feasible solution to TLOP.

The TLOP framework in Equation (Equation 3) considers the hierarchical multiple objectives. We introduced the equal sum and minimum decision method to evaluate the fitness of individuals in NHDE. The evaluation procedures regarding the fitness η for individuals are expressed as follows:(9)$$\eta =\min\left\{\eta_{CABL},\eta_{CAHT}\right\}$$
where $\eta_{CABL}$ and $\eta_{CAHT}$ denote the fitness of CA model to BL and HT models, respectively, and they are evaluated using the hierarchical objectives as follows:(10)$$\eta_{CABL}=\eta_{MUBL}+\min\left\{\eta_{MUBL},\left(SR^{T,BL}+SR^{T_{rxn},BL}\right)/2\right\}$$
The $\eta_{CAHT}$ is calculated using a similar procedure. The first priority of the objectives is to employ the fuzzy equal operation to determine that the mutant flux change ratios are as close to the templates as possible. The membership grades, $\eta_{MUBL}$ and $\eta_{MUHT}$, for each fuzzy equal of logarithmic fold change are described in Appendix A.

## 3. Results and Discussion

### 3.1. Templates of Flux Patterns for Cancer and Normal Cells

The metabolic model of Recon 2.2 consists of 5324 species, 7785 reactions, and 1675 associated genes (Figure 1A). Until date, Recon 2.2 has not been employed to reconstruct a tissue-specific GSMN model. Overall, 12865 gene expression profiles for normal and tumorous human tissues were acquired from HPA in the first step of the proposed procedures (Figure 1A). Based on the protein expressions and literature survey (Figure 1B), 1365 (including 74 reactions from literature) and 721 high confidence reactions for normal and cancerous colorectal tissues, respectively, were retrieved. The CORDA algorithm was applied to reconstruct the HT and CA colorectal models. The HT model comprised 2591 species, 4034 reactions, and 1483 genes, and the CA model comprised 2404 species, 3725 reactions, and 1454 genes. Figure 4 presents the intersections of species, reactions, and genes for both models. Both models were merged into the BL model to investigate how the healthy cell could be metabolically reprogrammed to become cancerous. Thus, the BL model is the union set of HT and CA models and includes 2692 species, 4284 reactions, and 1539 genes. These three models were engaged in the FBA and UFD problems to compute flux-sum distributions. The templates of flux-sum alterations for CA to BL and HT models were computed using the $LFC_{m}$ defined in Equation (Equation 4).

Total 565 choke-point metabolites (metabolites involve in only one synthesis or degradation reaction) in the normal and cancerous metabolic networks were determined by topology analysis. Due to that different uptakes can lead to a bias of the computational results, same uptakes (Appendix A) for normal and cancerous metabolic networks were used in this study.

### 3.2. Inferred Oncogenes

The NHDE algorithm was applied to solve the TLOP, and the top 20 one-hit oncogenes were determined; their average similarity ratios are presented in Table 1. These genes participated in various metabolic pathways. The average similarity ratio (Ave. SR) for each mutant was greater than 98%, and the extent of similarity for flux change ratio (Ave. CR) was greater than 93%. The inferred results revealed that these mutants have a similar flux pattern to that of cancer cells. Surveys of PubMed and GeneCards indicated the inferred oncogenes are related to various carcinomas and diseases, as shown in Table 1, and among the results, seven oncogenes were observed in CRC. From the survey of GeneCards, we obtained 1360 and 852 genes associated with FAP and HNPCC, respectively, and there are 117 FAP-associated and 49 HNPCC-associated enzyme genes included in the reconstructed colorectal model (Appendix A). Three enzyme genes (*CAT*, *G6PD*, and *CDO1*) obtained from the NHDE algorithm are FAP-associated genes. Therefore, the results reveal that the inference optimization framework is a useful computer-aided oncogene detection system.

We also applied RNA-sequencing data of the healthy and cancerous colorectal tissues obtained from the TCGA database [24] on the analysis of the 20 inferred oncogenes. The RNA-sequencing data consist of 41 healthy samples and 478 cancer samples. A two-tailed *t*-test was used to analyze the RNA-sequencing data, and a *p*-value less than 0.05 (typically ≤ 0.05) is statistically significant. The result shows that 17 out of the 20 oncogenes are significant as shown in Table 1. The RNA-sequencing data obtained from the TCGA database can also be employed to build GSMNs for oncogene inference in the TLOP problem. The reconstructed GSMNs were used to evaluate the similarity ratio and the extent of similarity of the mutant flux patterns for the 20 oncogenes (Appendix A). We found that 19 out of the 20 oncogenes have high similarity level as those in Table 1. Moreover, GSMNs reconstruted by the CORDA and iMAT algorithms based on Recon 2.02 and Recon 3D general models were used to validate the 20 oncogenes. The computation results are shown in Appendix A for comparison and reveals that these 20 oncogenes could also have high similarity ratios in different GSMNs.

Catalase, encoded by *CAT*, is an essential antioxidant enzyme that catalyzes the decomposition of hydrogen peroxide (H_2_O_2_) to water and molecular oxygen. *CAT* catalyzes three decomposition reactions of H_2_O_2_ in the cytoplasm, peroxisome, and mitochondria of the GSMN model. Based on the computation results, we determined that a 78% downregulation of *CAT* could increase the production of reactive oxygen species (ROS) that exhibits carcinogenic effects [38,39]. We used the STRING database [68] to investigate how *CAT* is related to apoptosis, proliferation, and cell growth signaling pathways (Figure 5A). We observed that *CAT* was strongly related to the *TP53*, *mTOR*, *AKT1*, *MAPK1*, *EGFR*, *MYC*, *CDK8*, and *RAS* family; these are well-known carcinogenic genes. Notably, *CDK8* is a verified oncogene that induces aberrant activation of the canonical WNT/β-catenin pathway occurring in almost all CRCs and contributs to cell growth, invasion, and survival [69]. We determined that *CAT* was also linked to *CDK8* through *TP53* and *MYC*. Finally, the 20 inferred genes in Table 1 were employed to investigate their protein-protein interactions (PPIs), as shown in Figure 5B. *CAT* showed all characteristics of a housekeeping gene, in that it acts as a hub to which most dysregulated genes are linked and then connected to the protein signaling pathways [40]. We determined from the STRING survey that *CYBRD1*, *CDO1*, and *LIPC* were disjointed in the PPI. The enzymes connected to *CAT* primarily participate in the glycolysis, pentose phosphate, amino acid, and lipid pathways. For example, *G6PD*, *H6PD*, *G6PC3*, *GPI*, *GRHPR*, and *SLC37A4* are associated with the glycolysis pathway. *G6PD* is a cytoplasmic enzyme that catalyzes the first step of the pentose phosphate pathway, which plays a key role in producing NADPH and the ribose required to synthesize DNA. *G6PD* deficiency is among the genetic factors hypothesized to protect against colorectal carcinogenesis [48].

### 3.3. Performance of Enzyme Pseudo-Coding

Furthermore, we applied a reaction-based approach to determine oncogenes in order to illustrate the performance of the proposed pseudo-enzyme strategy. Six top-ranked one-hit reactions are presented in Table 2. The reaction PGI is catalyzed by *GPI*, and the reaction r0161 is regulated by *AGXT*, such results are identical to that in Table 1. On the basis of the gene association of Recon 2.2, we determined that the gene *RPIA* catalyzes two reactions (r0249 and RPI). However, the reaction-based approach only determined r0249 with average similarity ratio of 0.981 and extent of flux changes of 0.935. The proposed pseudo-enzyme strategy was then applied to regulate both reactions. An average similarity ratio of 0.955 and average flux change ratio of 0.771 were obtained, which were smaller than that obtained using the reaction-based approach. Following the posterior procedures, we observed that the gene *HMGCL* could obtain an identical result through the reaction-based approach, but it also catalyzed another reaction HMGLx and included the isozyme HMGCLL1. The gene *PRODH2* achieved the same result but catalyzed four reactions. Finally, *CAT* regulated three reactions (CATp, CATm, and r0010), but each reaction achieved different results.

### 3.4. Flux Variability Analysis

To avoid the solution bias of FBA used in the TLOP problem for evaluation of flux alterations, flux variability analysis (FVA) was applied to compute the minimum and maximum values of each metabolite for the normal model and the mutants. The flux ranges were compared between mutant and normal models, and each flux difference between mutant and normal models was classified into seven categories according to the definition presented in Appendix A. The number of metabolites in different categories for the template and all mutants are shown in Figure 6. The result shows that about 71% of the synthesis rates of metabolites are complete decrease and about 5.8% complete increase. Such complete flux variances for all mutants are about 97% similar to the template and the average similarity ratios are about 82% as shown in Appendix A. Furthermore, the choke-point metabolites in the normal and cancerous models were determined by topology analysis, and total 565 choke-point metabolites were determined. The flux variance patterns of the metabolites with gene association for the template and mutants are shown in Figure 7. Such choke-point metabolites can be used as essential candidates for discovering biomarkers, because the single flux of synthesis or degradation reaction for each choke-point metabolite is corresponding to the gene expression measurement. We observed that major deregulations occurred in lipids, nucleotides, carboxylic acids, organic acids, organic oxygen compounds, and organic heterocyclic compounds. Lipids consist of fatty acyls, glycerophospholipids, prenol lipids, sphingolipids, cholestane steroids, and steroidal glycosides. The fluxes of six choke-point metabolites (cardiolipin, 4beta-methylzymosterol-4alpha-carboxylic acid, 14-demethyllanosterol, (S)-2,3-epoxysqualene, desmosterol, and 7-dehydrodesmosterol) among lipids are complete increase, and the others are decrease.

## 4. Conclusions

This study introduced a tri-level optimization framework that incorporated protein expressions of normal and tumor tissues for inferring oncogenes. The protein expression data or RAN-sequencing data, input of the optimization framework, was acquired from public databases such as HPA or TCGA. To enable provision of personalized medical treatment, the input of the optimization framework can be replaced with a patient’s protein expressio data. The proposed TLOP is an NP-hard problem, and no commercial software programs could be used to solve the problem. The NHDE algorithm with hierarchical objectives was applied to solve the problem. Moreover, this study introduced enzyme pseudo-coding numbers for expressing gene association to avoid the performing posterior decision-making that is necessary for the reaction-based method. CRC was applied as the case study, and the NHDE algorithm inferred 20 top-ranked oncogenes that cause various carcinomas and diseases based on surveys of PubMed and GeneCards. Notably, the evidence illustrated that the inference optimization framework enables oncogene prediction.

## Figures and Tables

**Figure 1 metabolites-10-00016-f001:**
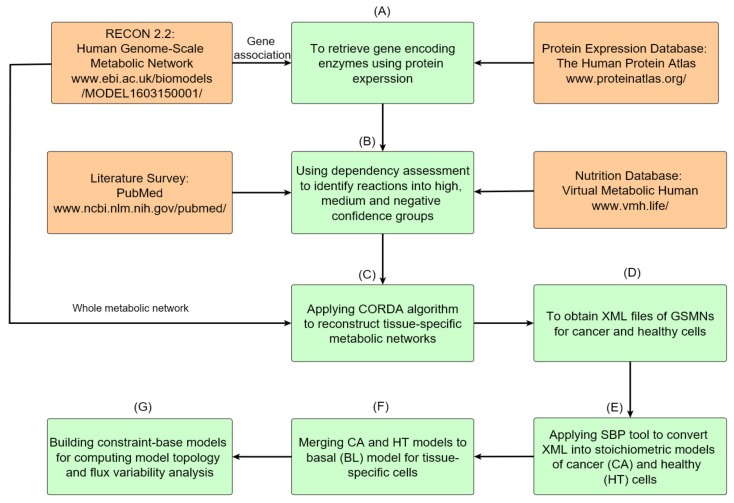
Roadmap of reconstruction of genome-scale metabolic networks for normal and cancer tissues. (**A**) Protein expressions of normal and cancerous colorectal tissues are accessed from HPA, and gene encoding enzymes are obtained through the gene association of Recon 2.2. (**B**) The acquired data are used to determine high, medium, and negative confidence sets of reactions. (**C**) The metabolic networks of cancer and healthy cells for the colorectal tissue are reconstructed using the CORDA algorithm. (**D**) Metabolic networks are stored in XML format. (**E**) The SBP tool is used to transfer metabolic networks to their stoichiometric models and gene-protein-reaction models. (**F**) Both cancer and healthy models are merged into a basal model. (**G**) The basal model can be used for further analysis and simulation.

**Figure 2 metabolites-10-00016-f002:**
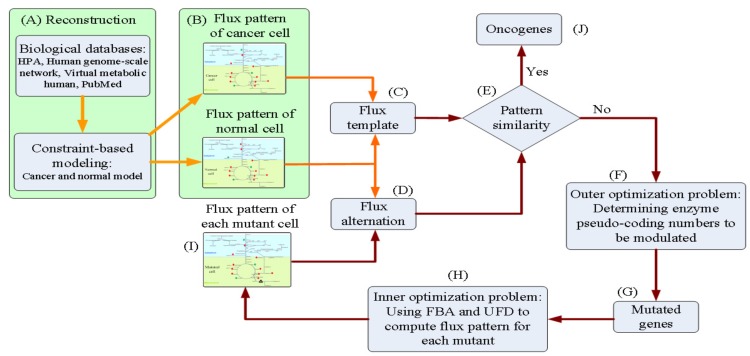
Flowchart of the in silico experiment for inferring oncogenes. (**A**) Reconstruct the cancer and normal models. (**B**) Compute the flux distributions of cancer and normal models. (**C**) Build the flux template according to the flux distributions of cancer and normal models. (**D**)–(**I**) Simulation of a wet lab experiment for determining oncogenes. Orange arrows indicate the building processes of the flux template that acted as the control in the oncogene inference problem. Red arrows present the mutant schemes for formulating the tri-level oncogene inference problem.

**Figure 3 metabolites-10-00016-f003:**
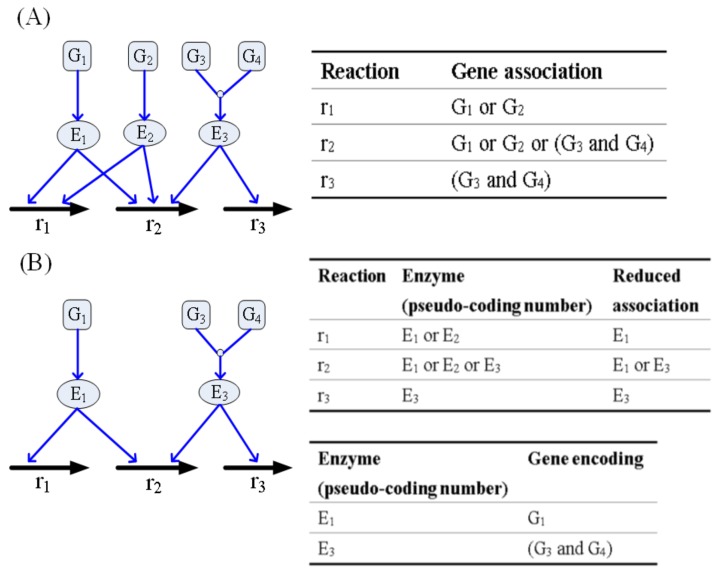
Example for building a gene-protein-reaction model using the enzyme pseudo-coding numbers. (**A**) Three reactions and their gene associations. (**B**) Reduced gene associations and encoded genes of the enzyme. E_2_ is a redundant enzyme that is identical to E_1_ and catalyzed the same reactions. r_2_ is catalyzed by the isozymes (E_1_ and E_3_). E_1_ is encoded by G_1_, and E_3_ is encoded by a complex of G_3_ and G_4_.

**Figure 4 metabolites-10-00016-f004:**
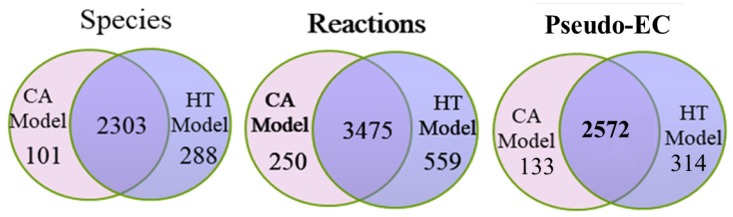
Statistics of cancer (CA) and healthy (HT) metabolomic models. Numbers of genes, species, and reactions for CA and HT models reconstruted by the CORDA algorithm taking the Recon 2.2 general model and HPA protein expression data as input. The basal (BL) model is composed of the union set of HT and CA models.

**Figure 5 metabolites-10-00016-f005:**
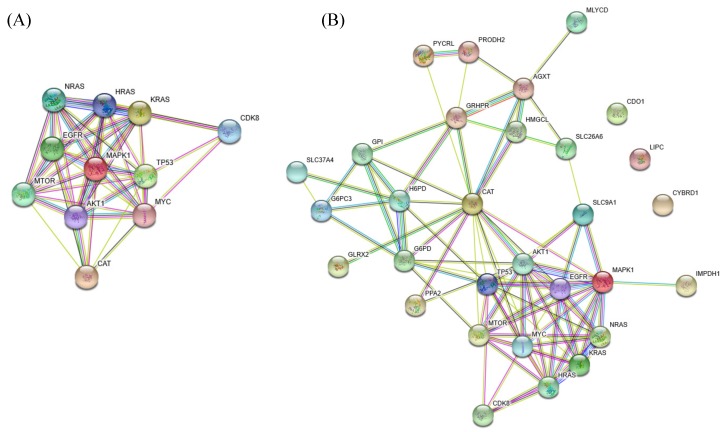
Protein-protein interactions (PPIs). (**A**) PPIs of the inferred oncogene *CAT*. *CAT* is strongly connected with the *TP53*, *mTOR*, *AKT1*, *MAPK1*, *EGFR*, *MYC*, *CDK8*, and *RAS* family. (**B**) *CAT* acts as a hub with most dysregulated genes linked to it.

**Figure 6 metabolites-10-00016-f006:**
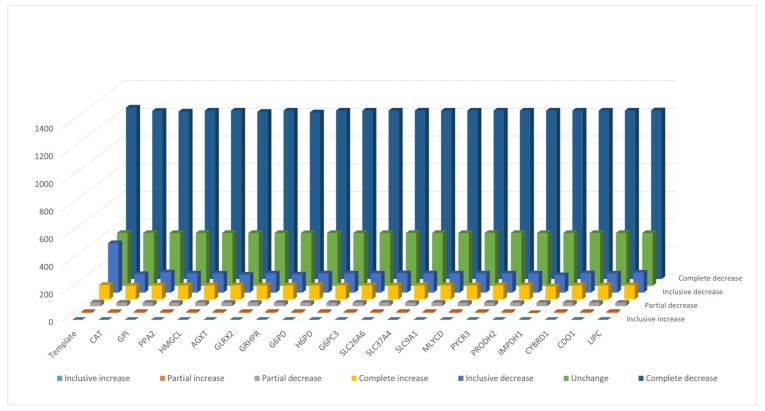
The number of metabolites in different categories for the template and 20 mutants. The definition of categories is presented in Appendix A.

**Figure 7 metabolites-10-00016-f007:**
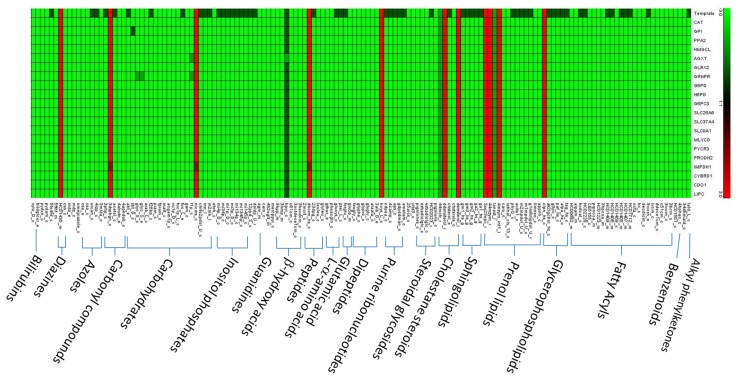
Flux variance patterns for the template and 20 mutants. Green indicates complete decrease, dark green means partial or inclusive decrease, and red denotes complete increase.

**Table 1 metabolites-10-00016-t001:** Top 20 one-hit oncogenes.

Gene	Pathway	Ave. CR ${}^{¶}$	Ave. SR ${}^{§}$	*p* Value	Disease (Score) ${}^{†}$	Remark ${}^{‡}$
*CAT*	Ethanol degradation	0.934	0.982	1.46 $\times 10^{-12}$	Gonadoblastoma (1.42) Amelanotic Melanoma (1.39)	Related to ROS signaling pathway [38,39,40].
*GPI*	Pentose phosphate pathway	0.931	0.981	6.57 $\times 10^{-35}$	Fibrosarcoma (1.08)	Gastric cancer [41].
*PPA2*	TRNA aminoacylation	0.935	0.982	0.4926	Sudden Cardiac Failure, Infantile (2.83)	Colorectal cancer [42] Prostate cancer [43].
*HMGCL*	Ketone body metabolism	0.935	0.982	3.8 $\times 10^{-29}$	3-Hydroxy-3-Methylglutaryl-Coa Lyase Deficiency (2.83)	Nasopharyngeal carcinoma [44].
*AGXT*	Alanine and aspartate metabolism	0.933	0.982	0.0133	Hyperoxaluria, Primary, Type I (2.83)	Colorectal cancer [45].
*GLRX2*	PAK pathway	0.932	0.982	4.35 $\times 10^{-9}$	NA	Oral squamous cell carcinoma [46].
*GRHPR*	Glyoxylate metabolism and glycine degradation	0.934	0.982	0.0018	Hyperoxaluria, Primary, Type Ii (2.83)	Hyperoxaluria [47].
*G6PD*	Methylene blue pathway	0.827	0.980	1.37 $\times 10^{-30}$	Anemia (2.63) Glutathione Synthetase Deficiency (1.50)	Colorectal cancer [48] Obesity and diabetes [49].
*H6PD*	Pentose phosphate pathway	0.918	0.982	0.0018	Cortisone Reductase Deficiency 1 (2.83)	Cancer cell lines for colon, breast and lung [50,51].
*G6PC3*	Carbohydrate digestion and absorption	0.936	0.982	4.55 $\times 10^{-53}$	Albinism, Oculocutaneous, Type Iv (1.26)	Breast cancer [52] Neutropenia [53].
*SLC26A6*	Mineral absorption	0.934	0.982	0.8577	Inflammatory Diarrhea (1.50)	Colorectal cancer cell lines [54] Pancreatic cancer cell [55].
*SLC37A4*	Carbohydrate digestion and absorption	0.930	0.982	0.4026	Glycogen Storage Disease (2.83) Pancreatic Ductal Adenocarcinoma (0.43)	Congenital hyperinsulinism of infancy [56].
*SLC9A1*	Osteoclast signaling	0.932	0.982	1.9 $\times 10^{-14}$	Lichtenstein-Knorr Syndrome (2.83) Breast Cancer (0.38)	Colon cancer cells [57] Gliomas [58].
*MLYCD*	Peroxisomal lipid metabolism	0.933	0.982	1.84 $\times 10^{-8}$	Malonyl-Coa Decarboxylase Deficiency (2.83) Pain-Chronic (1.43)	Proliferation of cancer cell lines [59].
*PYCR3*	Urea cycle and metabolism of amino groups	0.934	0.982	3.44 $\times 10^{-54}$	Lung Cancer Susceptibility (0.42)	Related to metastasis of cancer cells [60].
*PRODH2*	Arginine and proline metabolism	0.933	0.981	4.2 $\times 10^{-5}$	Primary Hyperoxaluria (1.34)	Hepatocellular carcinoma [61].
*IMPDH1*	Nucleotide metabolism	0.934	0.982	8.81 $\times 10^{-87}$	Leber Congenital Amaurosis (2.83)	Small cell lung cancer [62].
*CYBRD1*	Mineral absorption	0.934	0.981	0.0013	Iron Metabolism Disease (1.36)	Breast and prostate cancer cells [63].
*CDO1*	Taurine and hypotaurine metabolism	0.934	0.982	1.08 $\times 10^{-5}$	Small Intestine Cancer (1.31)	Colorectal cancer [64] Non-small cell lung cancer [65].
*LIPC*	Triacylglycerol degradation	0.940	0.981	0.0319	Hepatic Lipase Deficiency (2.83)	Colorectal cancer [66] Non-small cell lung carcinoma [67].

${}^{¶}$ Average similarity for flux change ratio; ${}^{§}$ Average similarity ratio of the mutant flux pattern to the templat; ${}^{†}$ Disease is obtained from GeneCards database and score is accessed from GeneCards; ${}^{‡}$ Brief description of gene function and references from PubMed and cancer databases.

**Table 2 metabolites-10-00016-t002:** Top six one-hit reactions. The criteria may overestimate or underestimate compared with the results solved by the pseudo-enzyme strategy.

Reaction	Gene	Other Regulated Reactions	Isozyme	Ave. CR ${}^{§}$	Ave. SR ${}^{†}$	Remark ${}^{‡}$
GPI	*GPI*	–	–	0.931	0.981	Gastric cancer [41].
r0161	*AGXT*	–	–	0.933	0.982	Colorectal cancer [45].
r0249	*RPIA*	RPI	–	0.935	0.981	Overestimated.
HMGLx	*HMGCL*	HMGLx	HMGCLL1	0.934	0.982	Nasopharyngeal carcinoma [44].
r0616	*PRODH2*	PROD2, r0615, PRO1x	–	0.934	0.982	Hepatocellular carcinoma [61].
CATp	*CAT*	CATPm, r0010	–	0.932	0.982	Related to ROS signaling [38,39,40].
CATm	*CAT*	CATp, r0010	–	0.838	0.979	Underestimated, ROS signaling [38,39,40].
r0010	*CAT*	CATm, CATp	–	0.867	0.981	Underestimated, ROS signaling [38,39,40].

${}^{§}$ Average similarity for flux change ratio; ${}^{†}$ Average similarity ratio of the mutant flux pattern to the template; ${}^{‡}$ Brief description of gene function and references from PubMed and cancer databases.

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
