# Peer review of "Genome-Scale Metabolic Modeling with Protein Expressions of Normal and Cancerous Colorectal Tissues for Oncogene Inference"

_metabolites, 2019, doi:10.3390/metabo10010016_

Round 1
Reviewer 1 Report
This work described the development of the tri-level optimisation (TLOP) framework to infer relevant oncogenes, using colorectal cancer as case study. The aim of using TLOP was to computationally mimic gene perturbations performed in wet-lab, and the infer oncogenes by “evolving” normal into cancer cells. Authors used Nested Hybrid Differential Evolution algorithm to solve this multi-level optimisation problem, to find mutants that have similar fluxes to cancer cells. Authors reconstructed tissue-specific normal and tumour models from Recon 2.2, guided using data from HPA, and introduced enzyme pseudo-coding number to help eliminate redundant gene-protein-enzyme mapping. Among the top 20 hits, Catalase was identified to be an important oncogene.
My major concerns for this work are the need for (i) more thorough validation of the oncogene hits to show the method has worked, and (ii) a clearer justification of the methodology. Hence, it is difficult to judge whether authors have been successful in their effort or not.
My ultimate concern is that authors findings could be largely driven by proteomics data. I did not find authors delivering their aim of finding more subtle oncogenes that increase susceptibility to CRC (line 66). Thus I feel more work is needed here.
More literature validation of hits to prove the utility of the developed methodology.
Authors results should recapitulate some of published findings that are specific for colorectal cancer, in particular, known metabolic re-wiring (doi:10.1038/s41416-019-0477-7) and clinical biomarkers (doi:10.1038/bjc.2016.243). Based my limited knowledge, Catalase do not appear to be a significant oncogene for colorectal cancer (the enzyme is broadly implicated in various cancers). Deregulation of lipid/sterol synthesis is a key metabolic characteristic of colorectal cancer; it is concerning this result (i.e., positive control) did not appear.
As control, I may suggest to authors to produce an additional case study (i.e., choose another cancer tissue) that have clearly different oncogenic drivers. This is to test whether the methodology can tell apart differences that are known.
Better justification and guidance to readers on the formulation of equations Did not show how values for wATP and wbiomass are established. These parameters would be important to contrast normal cells that have limited proliferation capacity versus tumours. This will ultimately affect fitness score and how NHDE select genes to modulate. It is not clear how “flux template” was built by “putting together flux pattern of normal and cancer cells”. Also, the purpose of a flux template was not clear to me. I am guessing authors wanted LFCs of mutant:normal to be similar to cancer:normal? UFD objective generally will select for the shorter pathway from a pair of redundant pathways. Is this why lipid synthesis pathways didn’t not appear among the hits? How did authors control uptake from extracellular, bearing in mind normal and cancer cells have different nutritional requirements? Formulation and purpose of “similarity ratios” difficult to follow.
Minor concerns
Reduce jargon in abstract. Hard to grasp value of work. Line 70. NFDE abbreviation first appear, needs full description. Line 54. I believe this study is not the first reconstruction of cancer and healthy cells and using topology to infer cancer drivers (reviewed here doi:10.15252/msb.20145307). Lacking in literature review of existing genome-scale model methodologies that compare normal and cancer cells/tissues. How is authors’ method better (i.e., the added value of TLOP framework)?
Reviewer 2 Report
The paper entitled « Genome-Scale Metabolic Modeling with Protein Expressions of Colorectal Normal and Cancer Tissues to Infer Oncogenes » by Feng-Sheng Wang et al. is divided into two parts. The first part describes reconstruction of tissue-specific genome-scale metabolic model of colorectal cancer and its healthy counterpart. In the second part Authors introduced a tri-level optimization problem (TLOP) simulating gene screen.
The manuscript is poorly written, especially in Abstract and Introduction sections. It would be nice to rewrite these two sections and even other parts.
Specific comments regarding to the first part:
Authors employed CORDA algorithm, as well as proteome data and Recon 2.2. It would be nice if they precisely mention what are the advantages of using these algorithm and data. Many algorithms have been developed to extract a tissue-specific GSMM, e.g. (https://www.cell.com/fulltext/S2405-4712(17)30010-8) and (https://journals.plos.org/ploscompbiol/article?id=10.1371/journal.pcbi.1003580). What is the advantage of using CODRA in this study? There are different platforms for production of omics-data, e.g., Microarray, RNA-Seq and Proteome-based. Meanwhile, we know that the Microarray and RNA-Seq technologies can measure expression of many genes rather than proteome-based technology. What is the advantage of using Proteome data in this study? Please mention the source of the data. Did you extract cell-line or patients’ data? Recon 3D, the last developed GSMM, consists of more than 13000 reactions and more than 3000 genes. Why Authors didn’t use of Recon 3D instead of Recon 2.2. Moreover, GSMM should not contain any block reaction and the original model should convert to a flux consistent model using ‘findFluxConsistentSubset’ function of COBRA toolbox(https://journals.plos.org/ploscompbiol/article?id=10.1371/journal.pcbi.1003424)
Authors of CORDA paper generated 20 cancer-specific models as well as the colorectal GSMM. It would be nice if Authors compare their reconstructed GSMM with the previous one by CORDA developers. It would be nice that Author mention which threshold they used for generating GSMMs and classified reactions into high, medium,76 and negative confidence groups. By using FBA, Authors measured the flux distribution of healthy and cancer models, but they used two different objective functions, was perhaps not the best idea?
Regarding to the second part, is there any other developed approaches in this area? If yes, please mention it and briefly describe the advantages of tri-level optimization problem (TLOP).
In Abstract Authors wrote “it is recently considered a metabolic disease” it is not true. The first discovery of metabolic alterations in cancer cells refers to Otto Warburg’s observations.
Round 2
Reviewer 2 Report
They addressed all my comments and I fully agree to be published.